# An “Instantaneous” Response of a Human Visual System to Hue: An EEG-Based Study

**DOI:** 10.3390/s22218484

**Published:** 2022-11-04

**Authors:** Gleb V. Tcheslavski, Maryam Vasefi

**Affiliations:** 1Drayer Department of Electrical Engineering, Lamar University, Beaumont, TX 77710, USA; 2Department of Biology, Lamar University, Beaumont, TX 77710, USA

**Keywords:** EEG analysis, EEG spatial vectors, response to solid color changing over time, similarity between multidimensional vectors, Kruskal–Wallis test

## Abstract

(1) The article presents a new technique to interpret biomedical data (EEG) to assess cortical responses to continuous color/hue variations. We propose an alternative approach to analyze EEG activity evoked by visual stimulation. This approach may augment the traditional VEP analysis. (2) Considering ensembles of EEG epochs as multidimensional spatial vectors evolving over time (rather than collections of time-domain signals) and evaluating the similarity between such vectors across different EEG epochs may result in a more accurate detection of colors that evoke greater responses of the visual system. To demonstrate its suitability, the developed analysis technique was applied to the EEG data that we previously collected from 19 participants with normal color vision, while exposing them to stimuli of continuously varying hue. (3) Orange/yellow and dark blue/violet colors generally aroused better-pronounced cortical responses. The selection of EEG channels allowed for assessing the activity that predominantly originates from specific cortical regions. With such channel selection, the strongest response to the hue was observed from Parieto-Temporal region of the right hemisphere. The statistical test—Kruskal–Wallis one-way analysis of variance—indicates that the distance evaluated for spatial EEG vectors at different post-stimulus latencies generally originate from different statistical distributions with a probability exceeding 99.9% (α = 0.001).

## 1. Introduction

Color is a powerful descriptor that is critically important for most animal species, as well as for humans. The ability to distinguish between approximately a million different colors (hues) allows us to attain a more accurate and detailed perception of the environment. Yet, the mechanisms of such perceptions are not completely understood, although recent evidence indicates that the visual cortex may be more sensitive to the hue of an object rather than to the color category [1]. Various neuroimaging modalities are used to assess neuronal activity and link it to visual perception. Among such modalities, electroencephalography (EEG) offers an excellent temporal resolution that allows studying neuronal activity that is time-locked to particular stimuli.

Numerous EEG analysis techniques have been applied to relate the cortical activity to the perceived color, including spectral estimates [2], Visual Evoked Potentials (VEPs) [3,4], fractal dimensions [5], EEG skewness [5,6], asymmetry [7], etc. Specifically, chromatic Visual Evoked Potentials (cVEPs) have been extensively studied during the last decades [8,9,10,11], while the particular features of cVEPs may be linked to specific participants’ conditions or even dysfunctions of their visual system [12,13].

We have reported previously that distinct local extrema may be observed in multivariate Temporal Response Functions (mTRFs) evaluated from the EEG of participants viewing the screen of changing color and these extrema should be attributed to the colors perceived by the participant rather than to the post-stimulus latencies. Therefore, we hypothesize that the specific color(s) may elicit a more pronounced cortical response. We have also suggested that specific EEG channels are responsible for better-defined results [14]. However, mTRFs appear to produce results prone to significant variation and, thus, lead to conclusions based on approximations. Therefore, the question of what specific color(s) may be linked to the more observable EEG features still remains. Additionally, mTRFs have been evaluated while using parametric modeling. As such, selecting an inappropriate model might lead to erroneous results.

The purpose of this study is to explore an alternative non-parametric EEG analysis technique capable of more accurate detection of stimulation-locked cortical activity and then to determine the specific colors evoking the most pronounced cortical response.

## 2. Materials and Methods

This study presents the verification of an alternative analysis technique applied to the color-evoked EEG data that we have reported previously [14]. We consider an ensemble of EEG epochs produced by “stacking” multichannel EEG fragments/epochs recorded as a response to the same stimulation. Since EEG is deemed related to neuronal activity and assuming that repeating the same stimulation should evoke similar responses, it is reasonable to expect that the EEG epochs forming such an ensemble should exhibit some common features, i.e., be “similar” in some sense. Traditionally, such ensembles of multidimensional vectors are processed, while considering their time dependence. Perhaps an alternative approach—viewing EEG as a spatial vector evolving over time—is also instrumental and thus may augment the traditional techniques.

### 2.1. Participants

This study was approved by the Lamar University IRB committee, and each participant has given informed consent. Nine male subjects aged between 19 and 30 (mean 22.9 years, standard deviation 2.6 years) and ten female subjects aged between 20 and 25 (mean 21.7 years, standard deviation 1.9 years) with normal color vision (verified by an Ishihara test [15]) have participated. Participants not passing the Ishihara test were referred to a specialist. The subjects reported no neurological problems or tiredness at the time of the EEG collection. Participants were seated in a comfortable chair in a dim environment with no direct lighting. Subjects were instructed to avoid body movements and eye blinking during the color stimuli presentation; blinking was allowed between the stimuli during the presentation of a black screen.

### 2.2. Visual Stimulation

Visual stimulation consisted of a video of a solid color that was changing over time. The intensity and saturation were kept approximately constant, while the hue (pure color) of the solid-color screen changed linearly from 0 to 1 (or from 0° to 360°). The rate of color variation was adjusted to allow the complete hue change cycle within approximately one, three, five, or fifteen seconds, resulting in video clips of the corresponding duration. These durations were selected empirically. Such short videos are referred to as “1-s”, “3-s”, etc. stimuli. The stimulation consisted of a randomized sequence of the abovementioned videos with a black screen presented between each stimulus for three seconds. The overall stimulation contained 15 “1-s” stimuli, 20 “3-s” stimuli, 10 “5-s” stimuli, and 5 “15-s” stimuli videos. The number of stimuli repetitions will be referred to as *K_s_*, where the index *s* is either 1, 3, 5, or 15. Therefore, *K_1_* = 15, *K_3_* = 20, *K_5_* = 10, and *K_15_* = 5. This combination of stimuli durations and the number of their repetitions was selected to maintain the overall experiment time under 10 min to reduce the participants’ fatigue.

Figure 1 illustrates the time course of the color presentation as the function of time for the four stimulation video clips. Assuming that the stimuli start at time “0” and for a specific post-stimulation time elapsed, the color observed on the screen would depend on the type of stimulus displayed. For instance, at 2 s (vertical blue line in Figure 1), the “15-s” stimulus would produce an orange screen, and the “5-s”—green, the ”3-s”—blue, and the “1-s” stimulus would be over, so the screen would be black.

Visual stimulation was presented using an overhead projector (Epson Home Cinema 8350) on a white screen located approximately two meters in front of the participants’ eyes. The image on the screen was approximately 2.2 m wide and 1.25 m high, effectively covering most of the subjects’ visual field with horizontal and vertical viewing angles of approximately 57 and 34 degrees, respectively.

### 2.3. EEG Acquisition

Electroencephalogram (EEG) was recorded using the ASA-Lab 40 system by ANT Neuro, Netherlands. Continuous EEG was pre-filtered in the 0.3–50 Hz range, notch-filtered at 60 Hz, sampled at 256 Hz, and recorded from 32 electrodes positioned according to the international 10/20 placement map. Stimuli presentation was synchronized with EEG acquisition by “eevoke”. Two identical trials (i.e., EEG recording attempts) were performed for most participants.

### 2.4. EEG Analysis: Pre-Processing and Segmentation

The EEG analysis was conducted using MATLAB. At the pre-processing step, DC components were removed from the EEG and the Common Average Reference spatial filter was applied to reduce surface currents. Next, each EEG data set was partitioned into *K_1_ + K_3_ + K_5_ + K_15_* = 50 epochs, synchronized with the visual stimulation on- and off-sets. A total of 50 epochs were extracted for each participant and each trial. The epochs were *N_1_* = 282, *N_3_* = 794, *N_5_* = 1289, and *N_15_* = 3775 sample-long, respectively.

Visual Evoked Potentials (VEPs) can be viewed as time-locked responses that may be stimulation-specific. Therefore, VEPs can be evaluated next by time-averaging over the *K_s_* EEG epochs corresponding to the same visual stimulation.

We have previously illustrated that an EEG epoch can be alternatively viewed as a sequence of “instantaneous” spatial vectors that evolve over time. This approach was shown to be instrumental for the VEP analysis [16] and appears suitable for studying the timing of stimulation-locked cortical activities with higher accuracy. Therefore, we will adopt the underlying concepts for the present study.

### 2.5. The “Instantaneous” Analysis

The proposed approach assumes that EEG can be viewed as an *M*-dimensional spatial vector (where *M* is the number of EEG channels determined by the hardware used) observed at the specific time instance, i.e., at the time when the EEG sample was collected for all EEG channels. Such spatial vectors can be compared with each other using a predefined similarity measure. Figure 2 illustrates the structure of the set of EEG epochs that correspond to the specific stimulus and the temporal vs. spatial vector representations. 

Brown/orange frames in Figure 2 illustrate the process of the epochs’ extraction/segmentation. For instance, “epoch 1” starts at the time index *n_0_* and is *N_s_*-sample long (*N_s_* is either 282, 794, 1289, or 3775). A “traditional”, i.e., the temporal vector, is shown by the blue dash/dot contour for EEG channel #2. This vector starts at the time instance *n_0_* and ends at *n_0_+N_s_*.

Alternatively, the *i*^th^ spatial (i.e., “instantaneous”) EEG vector is delimited by the green dash contour for all *M* EEG channels and corresponds to the time instance *n_i_*. The temporal vector concept can be used to estimate the VEPs. We suggest that the spatial vector representation “may be more adequate for the analysis of the perception-related alterations in the temporal distribution of the cortical activity” [16]. We also maintain that a spatial EEG vector can be viewed as an “instantaneous” snapshot of cortical activity taken from all available (or pre-selected) EEG channels.

From the conceptual viewpoint, each EEG epoch can be considered as a 2D array (a matrix) that contains *N_s_* by *M* elements. A juxtaposition of *K_s_*, such as the EEG epochs, can be viewed as a 3D array, **X**, of size *N_s_* by *M* by *K_s_*. The elements of **X** are X*_n,y,z_*, where *n* is the time variable that is in the range between *n*_0_ and *n*_0_
*+ N_s_* (Figure 2); *y* represents the EEG channel/electrode and varies from 1 to *M*; *z* is the epoch number that constitutes the set **X**. Here, *z* ranges between 1 and *K_s_*. Using the italic font to represent vectors and extending the abovementioned notation, for instance, the temporal vector representing EEG channel #2 from the first epoch (the blue dash/dot contour in Figure 2) can be denoted as *X_n,_*_2*,*1_; the *i*^th^ spatial/instantaneous vector in the first EEG epoch (the green dash contour in Figure 2) will be *X_i,y,_*_1_, etc.

The similarity between multi-dimensional vectors may be assessed with a number of techniques, including k-means clustering [17], multidimensional scaling [18], Pearson’s correlation coefficient [19], etc. For this study, we have adopted a generalized distance as the measure of similarity between two vectors as, perhaps, the most robust. Such a distance can be estimated as the *L^p^* norm of the difference between the coordinates of two arbitrary vectors, *Y* and *Z* [20]:(1)$$D_{p}={\left\|Y-Z\right\|}_{p}=\sqrt[{p}]{\sum_{j=1}^{M}{\left|Y_{j}-Z_{j}\right|}^{p}}$$

After removing the DC components and applying a spatial filter, EEG can be assumed as a zero-mean process. Therefore, we suggest that the absolute value should be included in (1) to potentially account for negative differences in the coordinates, i.e., the instantaneous voltages. Traditionally, the *L^2^* norm is utilized, which reduces (1) to the Euclidean distance. However, as the vectors’ dimensionality (the number of EEG channels, *M*, in our case) increases, lower orders or even fractional distance metrics may provide more meaningful results, especially for clustering applications [21]. Therefore, we will also assess fractional values of *p*.

Rather than conducting an exhaustive similarity assessment (i.e., each test vector with all other vectors in the epoch set), we will only compare the instantaneous vectors recorded at the same post-stimulus lag, as indicated by the red arrows in Figure 2. The latter reduces the number of necessary comparisons from the order of *N_s_^2^K_s_* to *N_s_K_s_*. Incorporating the notation for the spatial vectors introduced earlier, (1) can be rewritten for the post-stimulus latency *i* and between the epochs *a* and *b* as follows:(2)$$D_{p}{\left(a,b\right)}_{i}={\left\|X_{i,y,a}-X_{i,y,b}\right\|}_{p}=\sqrt[{p}]{\sum_{j=1}^{M}{\left|X_{i,j,a}-X_{i,j,b}\right|}^{p}}$$

The total number of unordered pairs of instantaneous vectors in the set of *K_s_* epochs while eliminating singletons (i.e., assuming that *a* ≠ *b*) is [22]:(3)$$L_{s}=\left(\begin{array}{l} K_{s}\\ 2 \end{array}\right)=\frac{K_{s}!}{\left(K_{s}-2\right)!2!}=\frac{K_{s}\left(K_{s}-1\right)}{2}$$

Combining (2) and (3), the average distance between these *L_s_* spatial vectors for the time latency *i* and all EEG epochs constituting the current epoch set is:(4)$${\bar{D}}_{p,i}=\frac{1}{L_{s}}\sum_{a\neq b}^{L_{s}}\sqrt[{p}]{\sum_{j=1}^{M}{\left|X_{i,j,a}-X_{i,j,b}\right|}^{p}}$$

We will be using the average distance (4) as a measure of similarity between the spatial vectors that correspond to the latency *i*.

Additionally, we can view the distance evaluated according to (2) as some metrics represented by a random variable with an unknown distribution. Thus, in addition to averaging, we may perform a statistical analysis (such as ANOVA or similar) to assess whether the distances evaluated at different post-stimulus latencies, say, *i* and *k*, originate from the same distribution or not. We selected a non-parametric test, the Kruskal–Wallis test, on ranks, which does not assume specific distributions of data. The distance evaluated at zero post-stimulation latency will contribute to one data sample, while the distance evaluated at another, non-zero latency, will constitute another sample used in the comparison. We can reasonably assume that the two data samples chosen in this manner are independent, since the metrics (i.e., the distances) are evaluated for different spatial vectors. The selection of the zero latency for the reference is arbitrary and can be revised without affecting the results. Details on the Kruskal–Wallis test can be found in [23].

## 3. Results

Figure 3 illustrates the Visual Evoked Potentials (VEPs) evaluated for one randomly selected participant and a single trial. These VEPs correspond to “1-s”, “3-s”, and “5-s” stimuli and were evaluated for the Oz, POz, and Cz electrodes, respectively. Similar to the previous study [14], we have observed that the “15-s” stimuli have not resulted in pronounced VEPs and thus were excluded from this report. Perhaps additional (other than the vision) physiological processes may contribute to the EEG while exposing subjects to prolonged visual stimulation.

The VEPs in Figure 3 were evaluated by averaging over 15, 20, and 10 EEG epochs, respectively. The VEPs are plotted as functions of the stimulation color, rather than time. As a result, for instance, the local maxima observed in Figure 3a,b, when the blue screen was projected correspond to different post-stimulation latencies for the “1-s”, “3-s”, and “5-s” stimuli, respectively. Therefore, we may argue that this particular VEP component may be a response to the specific stimulation color. On the other hand, the peaks observed for orange, yellow, and light-green colors for the “1-s” stimuli should, most likely, be attributed to the “pattern” VEP components, such as P100 and N145 [24] and, therefore, are not color-specific. The perhaps most studied component of VEPs, P300, may produce a peak at latencies between 250 and 700 ms [25]. Thus, we suggest that the portions of the VEPs observed during approximately the first ½ second following the stimuli onset may represent the combined result of two different perception mechanisms: the perception of color and the perception of light (i.e., the changing brightness). We further suggest that the classical, VEP-based processing does not appear as capable of detecting the colors producing stronger cortical responses by the continuous stimulation we have implemented. Additionally, the VEP-based approach appears to provide low temporal resolution, although it allows for the studying of the specific EEG channels, i.e., the electrical activity that predominantly originates from the specific cortical regions.

We maintain that the “instantaneous” spatial vector-based approach may be implemented as an alternative means of extracting the color-related information from EEG. Prior to applying this technique to the multi-channel EEG, its validity was evaluated by processing the synthetic data generated as a multi-channel pseudo-random normal process with a zero mean and the variance approximately matching that of the real EEG. The corresponding averaged distances (4) evaluated for both the synthetic data and real EEG of an arbitrarily selected subject are contrasted in Figure 4. Both results were averaged over 10 sets of either the real EEG or the synthetic data of the same dimensionality.

Consistent with our expectations, the distance plot evaluated for the synthetic data appears to randomly fluctuate around a seemingly arbitrary value, while the result evaluated from the EEG exhibits local minima and maxima for the specific latencies and generally produces lower values than those for the synthetic data. We suggest that this difference in appearance, to some extent, validates the proposed approach. Furthermore, the distance can be viewed as a measure of similarity: the lower distance may indicate that the underlying data vectors are similar to each other in some sense. Therefore, one may conclude that the “instantaneous” EEG vectors, constituting the same epoch at the latency of approximately 0.6s, may be similar to each other. Likewise, the vectors observed at the latency of approximately 4.3s may also be deemed similar, while the vectors at approximately 0.2s appear as the most dissimilar since the average distance between them reaches its maximum value.

The second norm was used when producing the results in Figure 4: i.e., *p* = 2 in (4). We have also evaluated the norms of the lower orders (i.e., 1 and 0.1). However, varying *p* resulted in changing the scale of the corresponding distance plots, while neither the plots’ general appearance nor the position of extrema seemed affected; therefore, the results for the norms of the lower orders are not included in this report.

Next, Figure 5 illustrates the average distance estimates, evaluated for all four stimuli and averaged over 19 participants, two trials per participant, and all corresponding epochs. A brief summary of the analyzed EEG epochs is provided in Table 1.

Zero on the time scale in Figure 5 corresponds to the stimulation onset, while the end of each graph is aligned with the stimulation offset for the corresponding stimuli; i.e., “1-s”, “3-s”, “5-s”, and “15-s”, correspondingly. All of the EEG channels, except for the mastoids, were used to produce the results. One may observe that every average distance graph exhibits distinct maxima and minima that do not appear as time-synchronized between the responses to different stimuli (except for the first minima for the “1-s”, “3-s”, and “5-s” stimuli). We may, perhaps, argue that the response during approximately the first second following the stimulation onset may predominantly represent the VEP, while the response after 1 s may be attributed mostly to color perception mechanisms. To assess this hypothesis, the average distance was next plotted as a function of observed color. In this case, as Figure 6 illustrates, the specific local extrema observed in the distance plots may be related to the perceived color.

The response to the “1-s” stimuli shows a single minimum for the green/teal color and appears dissimilar to the responses to the stimuli of the other durations. The responses to the “3-s” and “5-s” stimuli exhibit two distinct minima in the distance plots: around the orange/yellow and dark blue/violet colors. The teal/light-blue and red colors result in local maxima in the average distance. The response to the “15-s” stimuli appears more complex and with additional local extrema as compared to the stimulations with shorter stimuli. Perhaps, the latter may indicate that more complex perception mechanisms are involved. Yet, the pronounced minimum at the orange and a maximum at the teal colors are observed similarly to the shorter stimulations.

We maintain that, since distance (1) can be viewed as a measure of similarity, the local minima of the average distance plot indicate that the underlying spatial EEG vectors are “more similar” in some sense. Perhaps, this observation may suggest stronger responses of the visual system to the specific hues. Considering the “15-s” stimuli, the stronger responses are observed to orange, green, blue, and magenta colors. On the other hand, local maxima may suggest weaker responses of the visual system to other colors, such as red, yellow, and teal.

We have further observed that most female participants have produced distance maps that were somewhat inconsistent with similar maps evaluated for the male participants (Figure 7).

We observe in Figure 7 that the distance plots averaged for the nine male participants exhibit more consistent local minima/maxima and fewer variations than those evaluated for the ten female participants. The origin of these differences in the map appearance—whether they are due to some potential gender-specific differences in color perception or may be artifactual, for instance, related to a larger on average hair volume among the female participants (and thus poorer electrical contact)—would require an additional study. Not hypothesizing regarding the nature of such differences, we will only consider the results evaluated for male participants. We also conclude that the shortest, “1-s” stimuli seem to generate results that are generally inconsistent with those evoked by longer stimulations. Perhaps, this may be attributed to the experiment design that might lead to contaminating the response to colors by the response to the light intensity changes (i.e., light flashes). Therefore, we have excluded the results produced by the “1-s” stimuli in the rest of the report.

Applying this technique to the selected EEG channels may allow for the assessment of color processing by the specific cortical regions. Figure 8 illustrates the average distance evaluated for Occipital (O1, Oz, and O2), Parieto-Temporal (T7, T8, P7, P3, Pz, P4, and P8), and Frontal (Fp1, Fpz, Fp2, F7, F3, Fz, F4, and F8) regions averaged over the male participants and all the available trials. Occipital channels approximately cover the primary visual cortex, V1. Parieto-Temporal channels are located on top of area V4, which is traditionally associated with the processing of colors. Frontal channels primarily assess the activity of the Frontal lobe.

For the longest, i.e., “15-s”, stimuli, we observe in Figure 8 that the most pronounced response is registered from Parieto-Temporal region. Frontal region produces a well-defined, although weaker, response, and Occipital region generates the response that is the least defined.

Figure 9 illustrates, next, the average distance evaluated for Parieto-Temporal region, the left (a) and right (b) cerebral hemispheres.

We see in Figure 9 that the average distance evaluated for the right hemisphere shows less variation and better pronounced local extrema; therefore, the right hemisphere appears to produce a stronger response to color stimulation.

Before proceeding with the statistical analysis, it may be worth assessing the type of statistical distribution of the analyzed data. Figure 10 illustrates histograms evaluated for the distance estimated for the EEG spatial vectors corresponding to the “3-s”, “5-s”, and “15-s” stimuli for all the male participants, all the EEG electrodes, and all post-stimuli latencies.

As seen in Figure 10, the sample distributions appear to have long tails and certainly cannot be approximated by normal curves. Therefore, the non-parametric Kruskal–Wallis test should be preferred to the more traditional ANOVA. Figure 11 shows the H-values (red curve) evaluated between the distances estimated at the zero post-stimulation latency and all other latencies, the “3-s” stimuli. The blue curve illustrates the average distance and is identical to the corresponding curve shown in Figure 7a. The EEG data recorded from all the male participants and all the electrode locations were used.

We can draw two important conclusions based on Figure 11. Firstly, the average distance and H-values evaluated between the samples of the distance computed at different latencies appear to have opposite dynamics, i.e., when one curve has local minima, the other shows local maxima and vice versa. Most importantly, the majority of the H-values greatly exceed *H_c_*, the critical value, which, in our case, can be approximated by the χ^2^ distribution with one degree of freedom. Assuming the significance level α = 0.05, *H_c_* ≈ 3.84 for α = 0.001, the critical value increases to *H_c_* ≈ 10.83. Therefore, from Figure 11 (red curve) the null hypothesis—that the medians of the groups being compared are equal—should be rejected, even at the significance level of 0.001 (0.1%). In other words, we conclude that the distance between spatial EEG vectors is, in general, statistically different if evaluated at different post-stimulation latencies at the significance level of 0.001.

We further maintain that the maxima of the H-statistics should be attributed to the stimulation colors rather than the post-stimulation latencies. Figure 12 illustrates the H-values evaluated for the “3-s”, “5-s”, and “15-s” stimuli and presented as functions of the color perceived by participants.

We observe in Figure 12 that the H-statistics exhibit similar behavior when evaluated for the EEG data corresponding to different visual stimuli. More specifically, three curves show two local maxima corresponding to yellow and blue-violet stimulation colors. Perhaps, the difference in the scale between these curves may be attributed to the differences in the sample sizes. So, the sample size corresponding to the “3-s” stimuli was 3,420, while the sample sizes corresponding to the “5-s” and “15-s” stimuli were 810 and 180, respectively. Reducing the sample sizes to values of 50-100, for instance, by limiting the analysis to a single participant and one trial, scales the corresponding H-values down (not shown), while still exceeding the critical value for α = 0.001 at similar stimulation colors.

We should, however, note that the results of the statistical analysis (those depicted in Figure 10, Figure 11 and Figure 12) are only presented to further validate the utility of the spatial vector-based approach, rather than as a vital part of the technique.

## 4. Discussion

We conclude that using visual stimulation with dynamically varying color/hue allows for assessing the visual system responses to all available colors in the gamut. The latter constitutes the main advantage of the proposed stimulation approach compared to the stimulation with a finite set of fixed colors. Adjusting the speed of the hue variation allows varying for the temporal resolutions of the observed response.

We further conclude that the proposed “instantaneous” approach to the color-evoked visual response may be instrumental by providing an alternative view of the visual system response to color stimulation. The application of this technique to synthetic data can serve as a validation step. We maintain that considering ensembles of EEG epochs as multidimensional spatial vectors—rather than collections of time-domain signals—and evaluating the similarity between such vectors across different EEG epochs may result in a more accurate (compared to the traditional VEP approach) determination of colors that evoke stronger responses of the visual system. From Figure 6, we conclude that orange/yellow and dark blue/violet colors generally evoke more pronounced cortical responses.

The appropriate selection of the measure of similarity is an important aspect and should be the subject of additional investigations. In this study, we propose using the generalized distance between the spatial vectors as the measure of their similarity. Deeming the distance evaluated at specific latencies as random samples that originate from unknown statistical distribution(s), a Kruskal–Wallis analysis of the ranks indicates that such distances belong to distributions with different medians, with the probability exceeding 99.9% (the significance level of 0.001) when the distances are compared between the spatial EEG vectors recorded at different post-stimulation latencies. Our results also indicate that the statistical distribution of the distance cannot be approximated as normal.

Considering the longest (i.e., “15-s”) stimuli, the observation that Parieto-Temporal region produces the most pronounced color response (Figure 8b) is consistent with the current theory associating V4 with color processing [26,27,28]. This cortical area is a part of the ventral stream conveying the color information to the interior temporal lobe. Perhaps, the less defined color response from the occipital region (Figure 8a) may be explained by the “double-opponent” model [29,30], suggesting that regions of V1 respond best to the color contrast, rather than the colors themselves. Perhaps, shorter stimulation may be perceived as “more contrast”, thus eliciting a stronger response (Figure 8a). Additionally, a reasonably strong color response observed from Frontal region (Figure 8c) may indicate that color perception is also integrated into high-level cognitive processes, as suggested by other reports [31,32].

Considering Occipital region (Figure 8a), the longest “15-s” stimuli produce four well-pronounced minima, three of which correspond to orange, green, and blue colors. We may argue that these colors are somewhat similar to the colors/wavelengths of the peak sensitivity of the three types of cones, L, M, and S, in the human retina [33,34]. On the other hand, the L-cones (peaking at yellow/orange) are normally more numerous, while the S-cones (peaking at blue) are the least numerous in the human retina [35,36]. As seen in Figure 8a, the orange color evokes the strongest response, while the blue color elicits the weakest response. Perhaps, this response ranking may be correlated with the abundance of the corresponding cone cells constituting the retina. We may also notice that, although similar colors produce stronger responses in other cortical regions (i.e., the parieto-temporal and frontal, Figure 8b,c, respectively), the ranking of such responses is different than for those observed in the occipital region.

The lateral asymmetry in the response to the colors observed in Figure 9 appears consistent with the report by Bramao and colleagues [37] who suggested that the right V4 may carry more color processing than the left V4. Taylor and Xu [38] and Siuda-Krzywicka with co-workers [39] have recently reported somewhat similar observations, while the results of Singh and colleagues [40] may also be relevant.

A follow-up study assessing a larger and more diversified pool of participants is needed to verify the presented results and further justify the conclusions.

The spatial distribution of the response can be assessed while selecting the EEG channels included in the analysis. The proposed “instantaneous” approach may be further combined with time-domain techniques, such as VEP or mTRF analyses, to provide additional details of the spatial distribution of cortical activity.

On the other hand, alternating color stimuli (with an approximately constant “brightness”) and the black screen (with the “brightness” close to zero) may evoke the response to light during the first second following the color stimuli onset in addition to the response to the color itself. Therefore, the response to the color may be fused with the flash VEP, thus affecting the results. This possible contamination of the response to hue by the response to light is a potential limitation of the present stimulation that stems from the experiment design. To mitigate this, a grey image with an intensity matching the intensity of the color stimuli may be used instead of the black one.

Other stimuli modifications, such as stimuli reversal and/or non-linear hue variation, can be implemented to gain additional insight into the functionality of the visual system.

It may also be of interest to assess whether the specific features of the selected metrics (such as the extrema of the average distance or the H-values) may be subject-specific and, for instance, linked to individual color preferences, emotions experienced during the data acquisition, gender, age, cultural background, etc.

## 5. Conclusions

We conclude that the developed “instantaneous” approach to the analysis of vision-evoked cortical responses appears instrumental and may augment traditional techniques, such as VEPs. Although investigated (in the current study) for the specific case of color stimulation, we suggest that this approach may be easily extended to studying responses to other types of stimuli, such as visual, audible, tactile, etc.

From the experimental results of applying the proposed technique presented in this report, we made several observations that were generally aligned with the reports of other recent studies. The spatial vector-based EEG analysis appears capable of detecting cortical events synchronized with specific external stimuli. We have also discussed the potential limitations of the implemented stimulation protocol and underlined the follow-up study.

## Figures and Tables

**Figure 1 sensors-22-08484-f001:**
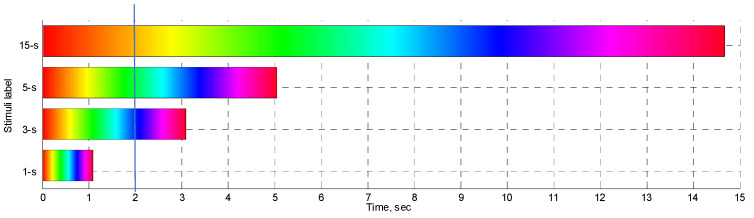
Time course of the color presentation for “1-s”, “3-s”, “5-s”, and “15-s” stimuli.

**Figure 2 sensors-22-08484-f002:**
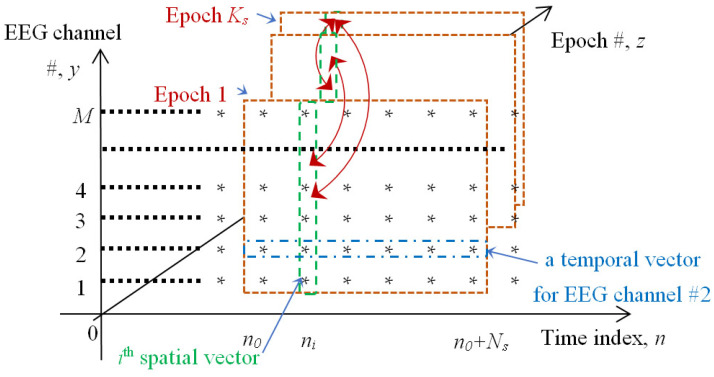
The structure of the set of EEG epochs corresponding to the specific stimulus.

**Figure 3 sensors-22-08484-f003:**
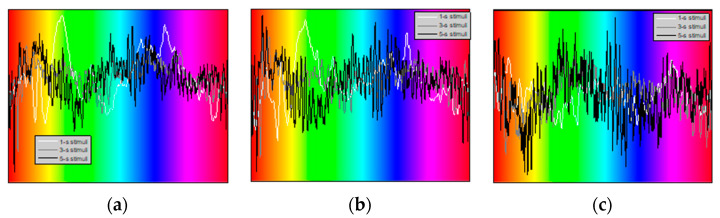
Visual evoked potentials for the “1-s”, “3-s”, and “5-s” stimuli and Oz (**a**), POz (**b**), and Cz (**c**) EEG channels.

**Figure 4 sensors-22-08484-f004:**
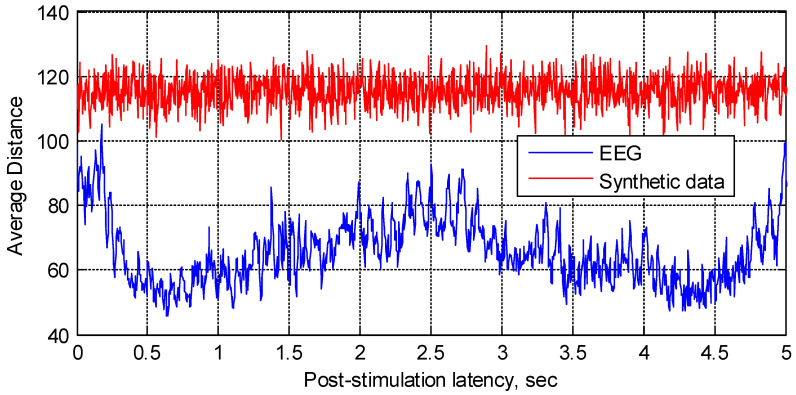
Distance averaged over 10 EEG epochs (one arbitrarily selected participant, single-trial, and “5-s” stimuli) and over 10 sets of synthetic data.

**Figure 5 sensors-22-08484-f005:**
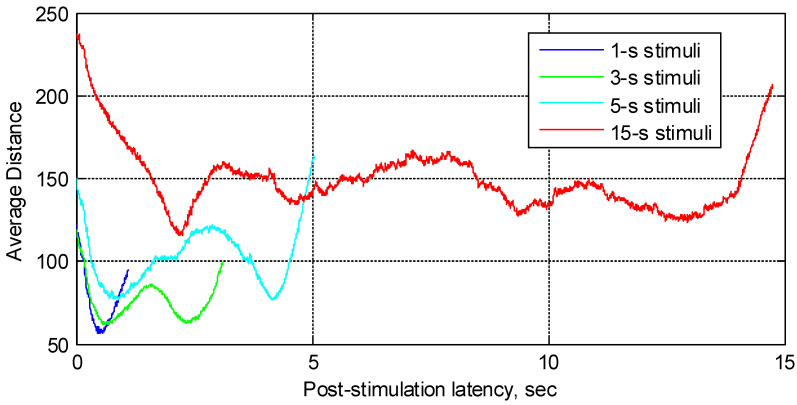
Distance averaged over 19 participants, 2 trials per participant, and all EEG channels and epochs as a function of post-stimulation latency.

**Figure 6 sensors-22-08484-f006:**
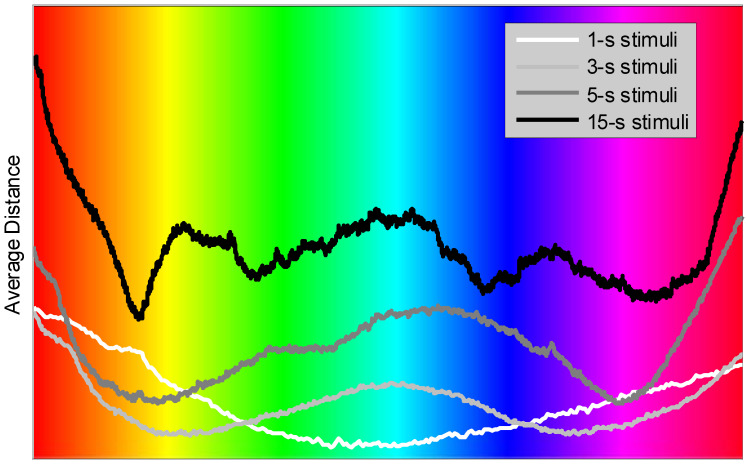
Distance averaged over 19 participants, 2 trials per participant, and all EEG channels as a function of stimulation color.

**Figure 7 sensors-22-08484-f007:**
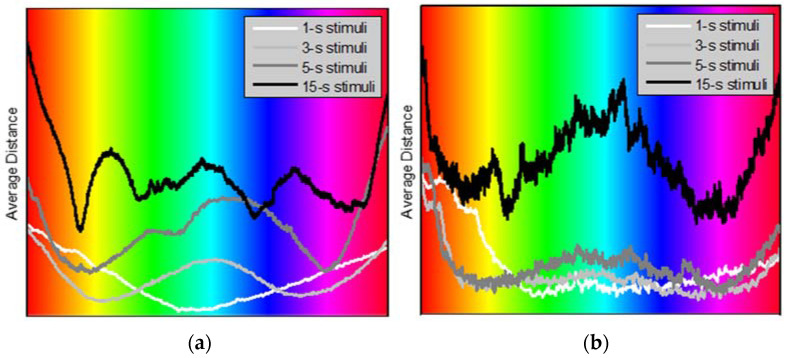
Average distance evaluated for male (**a**) and female (**b**) participants and for all EEG channels as a function of stimulation color.

**Figure 8 sensors-22-08484-f008:**
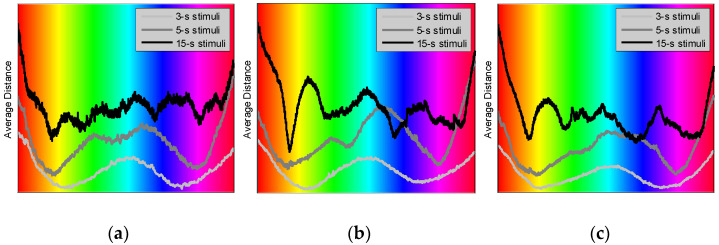
Average distance evaluated for the male participants and for Occipital (**a**), Parieto-Temporal (**b**), and Frontal (**c**) EEG channels as a function of stimulation color.

**Figure 9 sensors-22-08484-f009:**
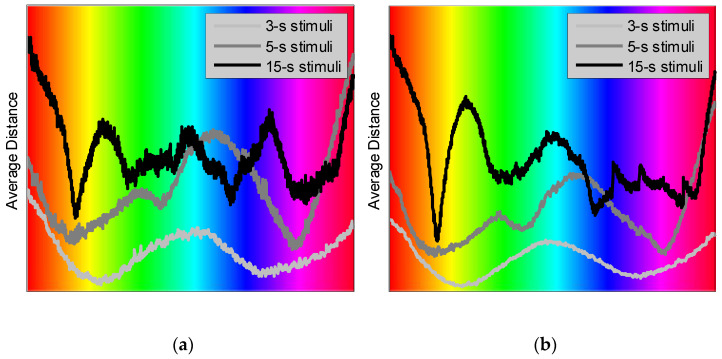
Average distance evaluated for the male participants and for the left (**a**) and right (**b**) Parieto-Temporal EEG channels as a function of stimulation color.

**Figure 10 sensors-22-08484-f010:**
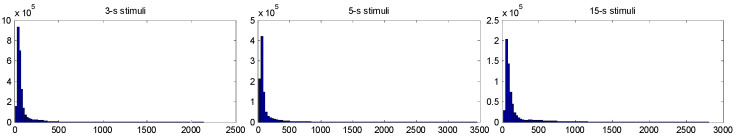
Histograms approximating the shape of sample distributions of the distance evaluated for spatial EEG vectors corresponding to the three color stimuli.

**Figure 11 sensors-22-08484-f011:**
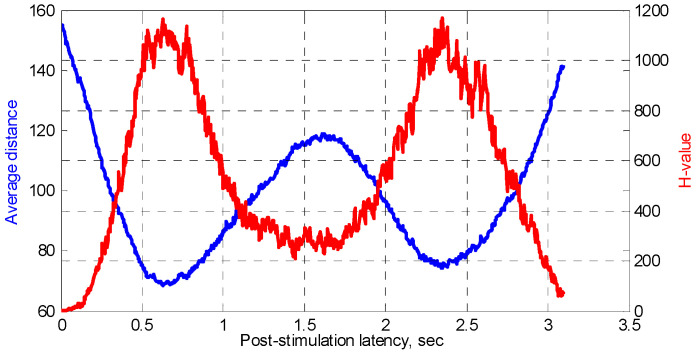
Average distance (blue) and H-values (red) evaluated for male participants and for all EEG channels as a function of post-stimulation latency.

**Figure 12 sensors-22-08484-f012:**
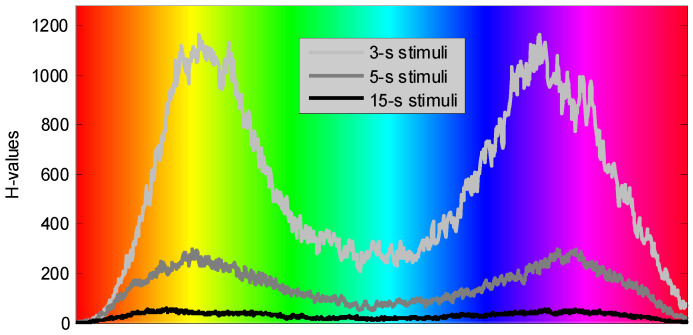
H-values evaluated for the male participants and for all EEG channels as functions of stimulation color.

**Table 1 sensors-22-08484-t001:** A summary of processed EEG epochs.

Stimuli	EEG Epoch Duration, Ns, Samples	Total Number of Available EEG Epochs
“1-s”	282	570
“3-s”	794	760
“5-s”	1289	380
“15-s”	3775	190

## Data Availability

Not applicable.

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
