# Peer review of "An “Instantaneous” Response of a Human Visual System to Hue: An EEG-Based Study"

_sensors, 2022, doi:10.3390/s22218484_

Round 1

Reviewer 1 Report

The present manuscript proposes a new approach to analyze the EEG visual evoked potential. In spite of a classic time-domain analysis, the authors propose to calculate the distance between spatial vectors of EEG value acquired at each time point in all electrodes at different epochs post-stimulus.

In my opinion, the manuscript is not suitable for publication in Sensors in the present form.

One of the main concern is that it is not clearly demonstrated the advantage of using this method instead of the classical time-domain analysis, since a time-locked response is looked for. A comparison between the results that one can obtain with the classical and the new approach proposed would have help the validation of the method. Moreover, in this new method, a temporal analysis seems to be required as well, since a statistical test had to be performed to verify the difference between the different time points after the stimulus.

It is not clearly elaborated in the manuscript the choice of the distance as metric of similarity. Pearson’s correlations or other similarity measures could have been more appropriate, if no other reasons are specified. In addition, it is not clear to the reader how to interpret a reduced distance, hence greater similarity, between vector of signal acquired at the same time-point after the stimulus and how this lack or increase in similarity is linked to the cerebral cortex response.

Another weakness reside in the experimental design: repeating two times the experiment for each subject and averaging all the measures seems to be a trick to increase the sample numerosity. However, these are not independent measures and this should be considered in the subsequent analysis. At the same time, presenting only the average distance results, without indication of the metric’s variance, does not allow the reader to actually identify the significant maxima/minima in the reported figure. Also, excluding all the female samples because of higher inter-subject variability, advocating the presence of artifactual components, seems a week argumentation. On the contrary, it can actually suggest a low robustness of the analysis with respect to the type of signal acquired.

In conclusion, in my opinion, in this work there is a lack of impacting novelty for the scientific community and there are some major issues about the validation method followed.

Author Response

Please, refer to the document attached.

Thank you!!

Reviewer 2 Report

The paper proposes An “instantaneous” response of a human visual system to hue: an EEG-based study. The article is well presented the methodological, experimental, and novelty of the paper after adding some recent literature work and removing grammatical mistakes then it could be considered for publication.

Author Response

(The authors gave the same response as above.)

Reviewer 3 Report

The authors propose a new technique to interpret biomedical data (EEG) to assess cortical responses to continuous color/hue variation.

The topic is interesting. The paper is well formatted. The language is good.

Technically: 

* The paper is not fortunate enough to be robust in terms of the number of samples used in the analysis.

* The time selection of video clips (1-s, 3-s, 5-s, and 15-s) is not justified. The K-i, also, are given empirically. Scientific justification is required.

 * The authors claimed “The purpose of this study is to explore an alternative non-parametric EEG analysis technique capable of more accurate detection of stimulation-locked cortical activity and to discriminate the specific colors evoking the most pronounced cortical response”, BUT, there is no single comparative result with a renowned technique.

 * The experimental setup should be explained more, by including the complexity of such proposed system.

Author Response

(The authors gave the same response as above.)

Round 2

Reviewer 1 Report

I accept the answers provided by the authors. I do not have further suggestions.

Author Response

Thank you very much for your excellent review and suggestions!

Reviewer 3 Report

The authors have improved the quality of their article by making it much clearer for non-specialist readers.

The paper has no conclusion. I recommend splitting the discussion section to show the conclusion in a separate section.

Author Response

Thank you very much for the excellent suggestion! We have added the following section to the manuscript:

  1. Conclusions

We conclude that the developed “instantaneous” approach to the analysis of vision-evoked cortical responses appears instrumental and may augment traditional techniques, such as VEPs. Although investigated (in the current study) for the specific case of color stimulation, we suggest that this approach may be easily extended to studying responses to other types of stimuli, such as visual, audible, tactile, etc.

From the experimental results of applying the proposed technique presented in this report, we made several observations generally aligned with reports of other recent studies. The spatial vector-based EEG analysis appears capable of detecting cortical events synchronized with specific external stimuli. We have also discussed the potential limitations of the implemented stimulation protocol and underlined the follow-up study.
